# Facile One-Pot Multicomponent Synthesis of Pyrazolo-Thiazole Substituted Pyridines with Potential Anti-Proliferative Activity: Synthesis, In Vitro and In Silico Studies

**DOI:** 10.3390/molecules26113103

**Published:** 2021-05-22

**Authors:** Islam H. El Azab, Rania B. Bakr, Nadia A. A. Elkanzi

**Affiliations:** 1Food Science and Nutrition Department, College of Science, Taif University, P.O. Box 11099, Taif 21944, Saudi Arabia; 2Department of Pharmaceutical Organic Chemistry, Faculty of Pharmacy, Beni-Suef University, Beni-Suef 62514, Egypt; rbbakr@ju.edu.sa; 3Chemistry Department, College of Science, Jouf University, P.O. Box 2014, Sakaka, Saudi Arabia; nahasan@ju.edu.sa; 4Chemistry Department, Faculty of Science, Aswan University, Aswan, P.O. Box 81528, Aswan, Egypt

**Keywords:** *N*-Heterocycles, multicomponent condensation, pyrazole-3-carbothioamide, thiazole, pyran, pyridine, anticancer activity, molecular docking

## Abstract

Pyrazolothiazole-substituted pyridine conjugates are an important class of heterocyclic compounds with an extensive variety of potential applications in the medicinal and pharmacological arenas. Therefore, herein, we describe an efficient and facile approach for the synthesis of novel pyrazolo-thiazolo-pyridine conjugate **4**, via multicomponent condensation. The latter compound was utilized as a base for the synthesis of two series of 15 novel pyrazolothiazole-based pyridine conjugates (**5**–**16**). The newly synthesized compounds were fully characterized using several spectroscopic methods (IR, NMR and MS) and elemental analyses. The anti-proliferative impact of the new synthesized compounds **5**–**13** and **16** was in vitro appraised towards three human cancer cell lines: human cervix (HeLa), human lung (NCI-H460) and human prostate (PC-3). Our outcomes regarding the anti-proliferative activities disclosed that all the tested compounds exhibited cytotoxic potential towards all the tested cell lines with IC_50_ = 17.50–61.05 µM, especially the naphthyridine derivative **7**, which exhibited the most cytotoxic potential towards the tested cell lines (IC_50_ = 14.62–17.50 µM) compared with the etoposide (IC50 = 13.34–17.15 µM). Moreover, an in silico docking simulation study was performed on the newly prepared compounds within topoisomerase II (3QX3), to suggest the binding mode of these compounds as anticancer candidates. The in silico docking results indicate that compound **7** was a promising lead anticancer compound which possesses high binding affinity toward topoisomerase II (3QX3) protein.

## 1. Introduction

The majority of the developed anticancer chemotherapies are not very effective, and side effects might concurrently occur, such as drug-induced impedance. Thus, there is still a critical need to develop novel effective and safe medicines with fewer side effects for the durable treatment of cancer [1,2]. Nitrogen-containing heterocyclic motifs are of high interest owing to their applications as pharmacologically active molecules. These molecules have gained cumulative attention, so they have contributed to the improvement of plentiful organic synthesis protocols and found ample applications in the chemical sciences [3,4,5,6,7]. Several *N*-heterocyclic conjugates are widely dispersed in nature and are constituents of many biologically important molecules, including several vitamins [8], antibiotics [9], nucleic acids [10], dyes and pharmaceuticals [11,12]. Moreover, the characteristics and utilization of *N*-heterocyclic skeletons (Figure 1) have gained a reputation in the rapidly growing fields of organic and therapeutic chemistry as well as the pharmaceutical industry [13,14,15]. On the other hand, the electron-donating heterocycle is not only able to readily receive or provide a proton, but it can also simply create various weak connections. Some of the intermolecular connections—for instance, hydrogen-bonding formation, van der Waals forces, dipole–dipole interactions, hydrophobic effects and π-stacking interactions—of *N*-heterocycles have amplified their significance in the field of therapeutic chemistry and allow them to efficiently adhere to a diversity of enzymes and receptors in drugs due to their improved solubility [16,17,18,19].

Among nitrogen heterocyclic analogues, pyrimidines have numerous applications in medicinal chemistry; the pyrimidine bases of uracil, cytosine and thymine are crucial building blocks of DNA and RNA [20]. In addition, pyrazole-containing scaffolds are a class of heterocycles that exhibit a wide range of biological effects, including anticancer [21], anti-HIV [22], antimalarial [23], anti-tubercular [24], anti-microbial [25] and diabetic activities [26,27]. 

Hybrid molecules containing thiazole scaffolds are a potential set of heterocyclic compounds; a thiazole core has been found in numerous biologically active natural drugs, such as thiamine (vitamin-B1), penicillin and luciferin [28]. There is a vast number of thiazoles that exhibit a wide range of pharmacological activities, including antibacterial [29], anticancer [30], antifungal [31], anti-inflammatory [32], antioxidant [33] and anti-tubercular activities [34]. Therefore, we believe that the merging of pyrazole and thiazole moieties will result in a class of pharmacophores that exhibit promising biological activities; in fact, several previous works have reported their potential activities, such as anticonvulsant, anti-HIV, anti-inflammatory and anticancer activities [35,36,37,38,39,40]. 

Based on the aforementioned applications of nitrogen-containing heterocycles, as well as the need for the construction of new bioactive *N*-heterocycles [41,42,43,44,45,46,47], herein, we report an efficient and facile approach for building up novel nitrogen-containing heterocycles with promising anti-proliferative effects, starting with 2-amino-6-(2-(1,5-dimethyl-1*H*-pyrazol-3-yl)-4-methylthiazol-5-yl)-4-(thiophen-2-yl)nicotinonitrile (**4**).

## 2. Results and Discussion

### 2.1. Chemistry

As an extension of our approach to the intended *N*-heterocycles [41,42,43,44,45,46,47], we studied the utilization of 2-(1,5-dimethyl-1*H*-pyrazol-3-yl)-5-acetyl-4-methyl-thiazole (**2**) to construct a potentially bioactive pyran and/or pyridine hybrids. Thus, coupling of the pyrazolecarbothiamide **1** with 3-chloropentane-2,4-dione in refluxing ethanol afforded the acetyl compound **2** in a 75% yield. Next, a multicomponent reaction of the acetyl compound **2** with thiophene-2-carbaldehyde and malononitrile under reflux in ethanolic piperidine solution yielded the pyran derivative **3**. Meanwhile, performing the same reaction with NH_4_OAc yielded the pyridine analogue **4** in an 81% yield (Scheme 1). A plausible mechanism for the construction of 2-amino-3-cyano pyridine moiety **4** using NH_4_OAc is displayed in Scheme 2, where the intermediate 2-(thiophen-2-ylmethylene)malononitrile **[A]**, which was obtained via the coupling of thiophene-2-carbaldehyde and malononitrile, was reacted with the thiazoylethenamine **[B]**; subsequently, an intramolecular cyclization, intermolecular rearrangement and auto oxidation yielded the pyridine analogue **4**. The spectral as well as the analytical data of **3** and **4** were consistent with their own structures (see Materials and Methods section).

Then, the reactivity of the *o*-aminonitrile tag in the pyridine analogue **4** was studied via cyclization with some active methylene compounds, to construct the envisioned pyridine and/or pyrrole nucleus-fused pyridine. Consequently, cyclization of the *o*-aminonitrile pyridine **4** under basic conditions in EtOH with diethylmalonate and malononitrile afforded 4-amino-1,8-naphthyridines (**5** and **6**) in fair yields (Scheme 3). Notably, the reaction product of compound **4** with ethyl cyanoacetate was extremely dependent on the reaction conditions. Thus, compound **4** refluxed with ethyl cyanoacetate in ethanolic piperidine solution yielded the 1,8-naphthyridine derivative **7** in a 73% yield, but when the same reaction was carried out under fusion, it yielded 2-cyanomethylpyrido[2,3-*d*]pyrimidin-4(3*H*)-one analogue (**8**) in an 82% yield. A reasonable mechanism for the formation of compound **8** is presented in Scheme 4. This reaction is assumed to proceed via an intermolecular nucleophilic attack of the amino pyridine analogy **4**, on the carbonyl carbon in the ethyl canoacetate, leading to the intermediate **4a**, and subsequent intramolecular cyclization via further, nucleophilic attack by the -OH moiety on the nitrile group afforded the cyanomethylpyrido[2,3-*d*][1,3]oxazine derivative **4b**. The intermediate **4b** undergoing intramolecular Dimroth-like rearrangement afforded the pyrido[2,3-*d*]pyrimidine analogue **4c**, and subsequent elimination of ethanol yielded the pyrido[2,3-*d*]pyrimidine analogue (**8**) (Scheme 4).

Moreover, fusing of a pyrrole nucleus with the pyridine moiety in compound **4** was achieved via the condensation of compound **4** with 2-chloroacetonitrile and/or ethyl bromoacetate in refluxing acetone and anhydrous K_2_CO_3_, which led to the new pyrrolo[2,3-*b*]pyridin-3-amine derivatives **9** and **10**, respectively (Scheme 3). The IR spectrum of compound **10** disclosed the lack of nitrile absorption that was initially observed in compound **4** (IR), while new absorption bands at 1664 and 3341 cm^−1^ were assigned to C=O and NH_2_ groups separately, and its mass spectrum showed an ion peak at *m*/*z* 478.12, which confirmed its molecular formula, C_23_H_22_N_6_O_2_S_2_. Moreover, the H-resonances of compound **10** showed triplet and quartet signals owing to the ethyl moiety at 1.26 and 4.32 ppm, respectively.

Next, the precursor **4** was subjected to further ring closure by coupling with some amino nucleophiles, which led to a set of pyridopyrimidine analogues (Scheme 5). In consideration of this, we treated compound **4** under reflux with formamide, hydroxylamine hydrochloride, thiourea, 6-methylpyridin-2-amine, 5-amino-3-phenyl-1*H*-pyrazole and/or 3-amino-1,2,4-triazole to construct pyridopyrimidines (**11**–**16**), in fair yields ranging from 65 to 91% (Scheme 5). The absence of nitrile absorbance was clearly observable in the IR spectra of these compounds. The mass spectrum of compound **16** exposed a molecular ion peak at *m*/*z* 459.12, which confirmed its molecular formula (C_21_H_17_N_9_S_2_) (see Appendix A).

### 2.2. Evaluation of Biological Impact

#### 2.2.1. Cytotoxic Activity

The newly constructed derivatives **5**–**13** and **16** were evaluated in vitro for their cytotoxic potential towards a prostate cancer cell line (PC-3), lung cancer cell line (NCI-H460) and cervical cancer cell line (Hela) using etoposide as a standard and adapting the MTT assay protocol [48,49]. The obtained outcomes (Table 1, Figure 2) showed that all the tested compounds exhibited cytotoxic potential against PC-3 (IC_50_ = 17.50–65.41 µM), NCI-H460 (IC_50_ = 15.42–61.05 µM) and Hela (IC_50_ = 14.62–59.24 µM) compared to etoposide (IC_50_ = 17.15, 14.28 and 13.34 µM, respectively). Moreover, 4-Amino-7-(2-(1,5-dimethyl-1*H*-pyrazol-3-yl)-4-methylthiazol-5-yl)-2-oxo-5-(thiophen-2-yl)-1,2-dihydro-1,8-naphthyridine-3-carbonitrile (**7**) was the most potent candidate towards PC-3 (IC_50_ = 17.50 µM), NCI-H460 (IC_50_ = 15.42 µM) and Hela (IC_50_ = 14.62 µM), which, in addition, revealed comparable anticancer potential to that exhibited by the standard drug etoposide (IC_50_ = 17.15, 14.28 and 13.34 µM, respectively). The pyridotriazolopyrimidin-5-amine analogy **16** was the least active candidate (IC_50_ = 65.41, 61.05 and 59.24 µM, respectively) towards the three tested cell lines, PC-30, NCI-H460 and Hela.

Anticancer activity was calculated using the MTT assay. Results are the average of three independent experiments run in triplicate.

#### 2.2.2. Structure–Activity Relationship

From the data obtained, it is clear that the naphthyridine derivatives **5**–**7** were the most active compounds against PC-3, NCI-H460 and Hela cell lines (IC_50_ = 17.5–29.31, 15.42–27.54 and 14.62–25.73, respectively). Within the naphthyridine derivatives, replacing the ethyl carboxylate moiety at C-3 as in compound **5** (IC_50_ = 25.73–29.31 µM) with the nitrile as in compound **6** (IC_50_ = 19.31–22.73 µM) and/or compound **7** (IC_50_ = 14.62–17.50 µM) markedly enhanced the anticancer potential. On the other hand, the presence of an amino group at C-2 of the naphthyridine ring as in compound **6** (IC_50_ = 19.31–22.73 µM) showed less potency than the 2-oxo moiety in compound **7** (IC_50_ = 14.62–17.50 µM) against all tested cell lines. Moreover, replacing the naphthyridine ring with the pyrrolo[2,3-*b*]pyridine system decreased the anticancer activity. This is clear upon comparing compound **7** (IC_50_ = 14.62–17.50 µM) with the pyrrolopyridine derivative **9** (IC_50_ = 31.05–35.12 µM). Furthermore, replacing the cyano group at C-2 of the pyrrole ring as in compound **9** (IC_50_ = 31.05–35.12 µM) with the carboxylate moiety as in compound **10** (IC_50_ = 45.62–48.29 µM) decreased the cytotoxic potency. Moreover, fusing the pyrimidine derivative **11** (IC_50_ = 58.12–59.13 µM) with the triazolo ring as in compound **16** (IC_50_ = 59.24–65.41 µM) reduced the anticancer potential against all tested cell lines.

### 2.3. Molecular Docking Study

To determine the mechanism of action behind the anticancer activity of the novel constructed compounds, these candidates were docked within topoisomerase II with the use of MOE software, 2010, version 8. An X-ray crystal of topoisomerase II with the cocrystallized ligand etoposide was attained from Protein Data Bank (code: 3QX3). Justification of the docking process was performed by redocking etoposide within topoisomerase II with RMSD = 0.9526. Etoposide formed two hydrogen bonds with AspB479 and DG C13 with binding score = −16.69 kcal/mol (Table 2).

The outcomes from this study illustrate that the novel compounds **5**–**13** and **16** fitted well within topoisomerase II. The most potent anticancer compound **7** recorded the highest binding energy score (−17.29 kcal/mol), showing two H-bond interactions with AspB479 with the carbonyl and amino groups of the pyridine ring. Moreover, the thiazole moiety of compound **7** recorded arene cation binding with ArgB503 (Figure 3).

On the other hand, compound **6** performed two types of interactions with the topoisomerase II active site. One is hydrophobic binding of the thiophen moiety with DGC13, and the other binding involves hydrogen bond interactions as follows: (i) AspB479 with NH_2_, and (ii) LysB456 with CN (Figure 4).

Moreover, compound **5** displayed hydrogen-bonding interactions with ArgB503 and DGC13 through binding with NH_2_ and C=O groups. Moreover, this naphthyridine derivative recorded hydrophobic binding with a thiophen moiety with DAC12 with binding score = −16.16 kcal/mol (Figure 5).

Moreover, the pyrimidine derivative 8 recorded binding energy score = −15.02 kcal/mol, forming three hydrogen bonds as follows (Figure 6):(i)DCC11 with carbonyl group;(ii)DCC11 with amino group;(iii)DGC10 with cyano group.

On the other hand, the cyano group of compound 9 formed a hydrogen bond with ArgB820, with a binding score equal to −17.32 (Figure 7).

The pyrrole moiety of pyrrolopyridine derivative 10 performed arene cation binding with ArgA820 with binding score = −14.06 kcal/mol. In addition, it formed two H-bonds with ArgB503 and DGC13 (Figure 8).

Furthermore, the pyrrole derivative **12** formed two H-bonds with ArgB820 and SerB818 through the nitrogen atom of the pyrrole ring and the amino group with binding score = −14.22 kcal/mol (Figure 9).

Compound **11** recorded a hydrophobic interaction with DAC12 and one hydrogen bond with AlaB816 (Figure 10).

Moreover, the pyridine ring of candidate **13** showed a hydrophobic interaction with DAC12 in addition to forming only one H-bond with DGC13 (Figure 11).

Finally, the least potent cytotoxic agent 16 formed only a hydrophobic interaction between the thiophen ring and DAC12, without displaying any hydrogen bond (Figure 12).

## 3. Materials and Methods

### 3.1. General Description of Materials and Methods

The chemicals used in this work were obtained from Sigma Aldrich (Palo Alto, CA, USA) and were used without any further purification.

### 3.2. Instrumentation

All the synthesized compounds were elucidated by NMR (^1^H, ^13^C), MS and IR (see Appendix A). IR data were documented as KBr discs utilizing a Bruker-Vector 22 FTIR spectrophotometer (Bruker, Manasquan, NJ, USA). The NMR spectra were verified with a Varian Mercury VXR-300 (Bruker, Marietta, GA, USA), at 300 and 75 MHz (^1^H and ^13^C-NMR) spectra separately, as a solution in DMSO-*d_6_*. The chemical shifts are presented in *δ* scale relative to the internal reference tetramethylsilane (TMS). Mass spectra were recorded at 70 eV on a Hewlett Packard MS-5988 spectrometer (Hewlett Packard, Palo Alto, CA, USA). Elemental analyses were conducted at the Micro-Analytical Center of Taif University, Taif, Saudi Arabia.

### 3.3. Synthetic Procedures and Analytic Data of Compounds

*1-(2-(1,5-Dimethyl-1H-pyrazol-3-yl)-4-methylthiazol-5-yl)ethanone* (**2**). In an ethanolic solution (40 mL), the pyrazole carbothiamide **1** (0.15 g, 1 mmol) and 3-chloro-2,4-pentanedione (1 mmol) were boiled at reflux for 7 h (tested by TLC). On cooling at ambient temperature, the obtained solid was filtered, washed with cold methanol and recrystallized using EtOH to furnish the acetyl thiazole analogy **2**, as a pale yellow solid. Yield: 75%; m.p. 242–244 °C; IR (KBr): (cm^−^^1^) 683 (C–S–-C), 1610–1625 (2C=N), 1730 (C=O); ^1^H-NMR: δ 1.39, 2.02, 3.25 (s, 9H, 3Me), 3.12 (s, 3H, Ac), 6.62 (s, 1H, Pyraz._(C4)_-H); ^13^C-NMR: 10.7, 15.8, 27.4, 38.3 (4Me), 106.8, 138.3, 143.4, 161.2 (2C=C), 136.3, 162.8 (2C=N), 196.5 (C=O); MS (*m*/*z*, %): 235.09 (M^+^, 15); Anal. Calcd. for C_11_H_13_N_3_OS (235.31): C, 56.15; H, 5.57; N, 17.86%. Found: C, 56.01; H, 5.32; N, 17.67%.

*2-Amino-6-(2-(1,5-dimethyl-1H-pyrazol-3-yl)-4-methylthiazol-5-yl)-4-(thiophen-2-yl)-4H-pyran-3-carbonitrile* (**3**). In a mixture of dry ethanol, piperidine (25 mL:0.5 mL), the acetyl thiazole **2** (0.23 g, 1 mmol), thiophene-2-carbaldehyde (0.12 g, 1 mmol) and malononitrile (0.06 g, 1 mmol) were mixed and refluxed for 3 h (examined by TLC). After cooling the reaction mixture to RT, it was transferred onto ice/H_2_O and neutralized by HCl, and the solid separated out was isolated, splashed with H_2_O, dried out and recrystallized by EtOH to yield compound **3** as a buff powder. Yield: 77%; m.p. 172–174 °C; IR (KBr): (cm^−1^) 675–681 (2C–S–C), 1610–1625 (2C=N), 2211 (C≡N), 3281 (NH_2_); ^1^H-NMR: *δ* 2.42, 2.45, 3.64 (s, 9H, 3Me), 3.96 (d, 1H, *J* = 8.5, Pyran._(C4)_-H), 5.01 (d, H, *J* = 8.5, Pyran._(C5)_-H), 6.01 (s, 1H, Pyraz._(C4)_-H), 6.51 (brs, 2H, NH_2 Deutr. Exch_), 6.83–7.49 (m, 3H, Thioph.-H); ^13^C-NMR: 10.7, 15.8, 38.3 (3Me), 29.8 (pyran C-4), 119.1 (C≡N), 58.1, 97.5, 106.3, 123.4, 125.5, 127.0, 138.2, 139.7, 143.0, 153.9, 159.2, 161.0 (6C=C), 136.3, 162.8 (2C=N); MS (*m*/*z*, %): 395.09 (M^+^, 20); Anal. Calcd. for C_19_H_17_N_5_OS_2_ (395.50): C, 57.70; H, 4.33; N, 17.71%. Found: C, 57.42; H, 4.15; N, 17.51%.

*2-Amino-6-(2-(1,5-dimethyl-1H-pyrazol-3-yl)-4-methylthiazol-5-yl)-4-(thiophen-2-yl)nicotinonitrile* (**4**). To a solution of the acetyl compound **2** (0.23 g, 1 mmol) in a mixture of dry EtOH (40 mL) containing AcONH_4_ (0.53 g, 7 mmol), thiophene-2-carbaldehyde (0.12 g, 1 mmol) and malononitrile (0.06 g, 1 mmol) were added and mixed together. The reaction mixture was heated at reflux for 3 h (examined by TLC), then allowed to cool to RT, transferred onto mashed ice and neutralized by HCl, and the solid separated out was collected, splashed with H_2_O, dried out and recrystallized using EtOH to yield compound **4** as a yellow powder. Yield: 81%; m.p. 158–160 °C; IR (KBr): (cm^−1^) 675–681 (2C–S–C), 1610–1625 (2C=N), 2211 (C≡N), 3281 (NH_2_); ^1^H-NMR: *δ* 2.26, 2.40, 2.46 (s, 9H, 3Me), 6.06 (s, 1H, Pyraz._(C4)_-H), 6.49 (brs, 2H, NH_2 Deutr. Exch_), 7.23–7.36 (m, 4H, Thioph.-H and Pyrid._(C5)_-H); ^13^C-NMR: 10.7, 13.8, 38.3 (3Me), 113.7 (C≡N), 86.3, 106.3, 111.7, 126.8, 127.2, 128.2, 128.6, 138.2, 138.5, 148.4, 153.3, 162.2 (6C=C), 136.3, 155.7, 162.8 (3C=N); MS (*m*/*z*, %): 392.09 (M^+^, 40); Anal. Calcd. for C_19_H_16_N_6_S_2_ (392.50): C, 58.14; H, 4.11; N, 21.41%. Found: C, 58.01; H, 4.02; N, 21.22%.

*General method for the synthesis of compounds* (**5**–**7**). To a solution of pyridine analogy **4** (0.39 g, 1 mmol), in a mixture of ethanol, piperidine (25 mL: 0.3 mL), some selected carbon donors—namely diethylmalonate, malononitrile and ethyl cyanoacetate—were added. Next, the reaction mixture was refluxed for 4–6 h (as tested by TLC), and the solvent was extracted under reduced pressure. The obtained solid was collected, dried and recrystallized by the proper solvent, providing the desired compounds (**5**–**7**).

*Ethyl 4-amino-7-(2-(1,5-dimethyl-1H-pyrazol-3-yl)-4-methylthiazol-5-yl)-2-oxo-5-(thiophen-2-yl)-1,2-dihydro-1,8-naphthyridine-3-carboxylate* (**5**). After recrystallization using AcOH, compound **5** was attained as yellow crystals. Yield: 79%; m.p. 216–218 °C; IR (KBr): (cm^−1^) 675–681 (2C–S–C), 1610–1625 (3C=N), 1648 (amidic C=O), 1729 (ester C=O), 3320–3350 (NH and NH_2_); ^1^H-NMR: *δ* 1.29 (t, 3H, *J* = 6.5, Me), 2.42, 2.45, 3.64 (s, 9H, 3Me), 3.06 (q, 2H, *J* = 6.5, CH_2_), 6.01 (s, 1H, Pyraz._(C4)_-H), 6.40 (brs, 2H, NH_2 Deutr. Exch_), 6.83–7.49 (m, 4H, Thioph.-H and Pyrid._(C5)_-H), 10.21 (brs, H, NH _Deutr. Exch_); ^13^C-NMR: 10.8, 12.5, 14.2, 38.0 (4Me), 61.0 (CH_2_), 86.2, 103.1, 106.3, 111.7, 127.6, 127.9, 128.0, 128.6, 138.2, 138.5, 148.6, 152.4, 162.0, 169.6 (7C=C), 136.3, 155.6, 162.8 (3C=N), 161.1, 165.0 (2C=O); MS (*m*/*z*, %): 506.12 (M^+^, 31); Anal. Calcd. for C_24_H_22_N_6_O_3_S_2_ (506.60): C, 56.90; H, 4.38; N, 16.59%. Found: C, 56.61; H, 4.12; N, 16.29%.

*2,4-Diamino-7-(2-(1,5-dimethyl-1H-pyrazol-3-yl)-4-methylthiazol-5-yl)-5-(thiophen-2-yl)-1,8-naphthyridine-3-carbonitrile* (**6**). After recrystallization using ethanol–DMF (3:1), compound 6 was obtained as a yellow powder. Yield: 71%; m.p. 189–191 °C; IR (KBr): (cm^−1^) 675–681 (2C–S–C), 1610–1625 (4C=N), 2210 (C≡N), 3320–3350 (2NH_2_); ^1^H-NMR: δ 2.26, 2.40, 2.46 (s, 9H, 3Me), 6.06 (s, 1H, Pyraz._(C4)_-H), 5.35, 6.49 (brs, 4H, 2NH_2 Deutr. Exch_), 7.23–7.36 (m, 4H, Thioph.-H and Pyrid._(C5)_-H); ^13^C-NMR: 10.7, 13.8, 38.3 (3Me), 113.7 (C≡N), 77.8, 86.3, 106.8, 124.3, 126.8, 127.2, 128.2, 128.6, 138.2, 138.5, 148.4, 155.3, 158.3, 162.0 (7C=C), 136.3, 155.7, 162.2, 162.8 (4C=N); MS (*m*/*z*, %): 458.04 (M^+^, 47); Anal. Calcd. for C_22_H_18_N_8_S_2_ (458.56): C, 57.62; H, 3.96; N, 24.44%. Found: C, 57.41; H, 3.68; N, 24.30%.

*4-Amino-7-(2-(1,5-dimethyl-1H-pyrazol-3-yl)-4-methylthiazol-5-yl)-2-oxo-5-(thiophen-2-yl)-1,2-dihydro-1,8-naphthyridine-3-carbonitrile* (**7**). After recrystallization using DMF–MeOH (1:3), compound **7** was obtained as brownish-yellow fine grains. Yield: 73%; m.p. 159–161 °C; IR (KBr): (cm^−1^) 675–681 (2C–S–C), 1610–1625 (3C=N), 2210 (C≡N), 3280 (NH_2_); ^1^H-NMR: *δ* 2.42, 2.45, 3.64 (s, 9H, 3Me), 5.62 (brs, 2H, NH_2 Deutr. Exch_), 6.01 (s, 1H, Pyraz._(C4)_-H), 6.83–7.49 (m, 4H, Thioph.-H and Pyrid._(C5)_-H), 10.12 (brs, 1H, NH _Deutr. Exch_); ^13^C-NMR: 10.8, 12.5, 38.0 (3Me), 115.8 (C≡N), 80.7, 106.3, 111.1, 112.3, 127.6, 128.0, 128.6, 138.2, 138.4, 144.2, 149.8, 176.8 (6C=C), 136.3, 151.3, 162.8 (3C=N), 168.3 (C=O); MS (*m*/*z*, %): 459.10 (M^+^, 10); Anal. Calcd. for C_22_H_17_N_7_OS_2_ (459.55): C, 57.50; H, 3.73; N, 21.34%. Found: C, 57.29; H, 3.41; N, 21.15%.

*2-(7-(2-(1,5-Dimethyl-1H-pyrazol-3-yl)-4-methylthiazol-5-yl)-4-oxo-5-(thiophen-2-yl)-3,4-dihydropyrido [2,3-d]pyrimidin-2-yl)acetonitrile* (**8**). Without any solvent, the pyridine analogy **4** (0.39 g, 1 mmol) and ethyl cyanoacetate (10 mL) were fused for 5 h. After cooling to RT, the reaction mixture was triturated with cold ethanol, and the separated solid was extracted and recrystallized by EtOH, to provide the cyanomethyl pyrimidine analogy **8**, in an 82% yield; m.p. 253–256 °C; IR (KBr): (cm^−1^) 675–681 (2C–S–C), 1610–1625 (4C=N), 1645 (C=O amide), 2210 (C≡N), 3280 (NH); ^1^H-NMR: *δ* 2.42, 2.45, 3.64 (s, 9H, 3Me), 4.12 (s, 2H, CH_2_), 6.01 (s, 1H, Pyraz._(C4)_-H), 6.83–7.49 (m, 4H, Thioph.-H and Pyrid._(C5)_-H), 10.12 (brs, 1H, NH _Deutr. Exch_); ^13^C-NMR: 10.8, 12.5, 38.0 (3Me), 22.4 (CH_2_), 116.3 (C≡N), 106.3, 118.5, 121.5, 127.6, 127.9, 128.0, 128.6, 138.2, 138.4, 147.5, 152.4, 154.0 (6C=C), 136.3, 152.1, 156.4, 162.8 (4C=N), 161.0 (C=O); MS (*m*/*z*, %): 459.09 (M^+^, 41); Anal. Calcd. for C_22_H_17_N_7_OS_2_ (459.55): C, 57.50; H, 3.73; N, 21.34%. Found: C, 57.29; H, 3.41; N, 21.15%.

*General Procedure for the Synthesis of Compounds* (**9** and **10**). To a solution of compound **4** (0.39 g, 1 mmol) and anhydrous K_2_CO_3_ (1.37 g) in dry acetone (25 mL), chloroacetonitrile and/or ethyl bromoacetate (20 mmol) was added. After, the reaction mixture was refluxed for 5 h (monitored by TLC), and the extra solvent was removed under reduced pressure. The resulted rough matter was triturated by cold MeOH, and the separated solid was filtered, washed, dried and recrystallized to give compounds **9** and **10**.

*3-Amino-6-(2-(1,5-dimethyl-1H-pyrazol-3-yl)-4-methylthiazol-5-yl)-4-(thiophen-2-yl)-1H-pyrrolo[2,3-b]pyridine-2-carbonitrile* (**9**). After recrystallization using methanol compound **9** obtained as an off-white solid. Yield: 78%; m.p. 261–263 °C; IR (KBr): (cm^−1^) 675–681 (2C–S–C), 1612–1625 (3C=N), 2208 (C≡N), 3320–3350 (NH and NH_2_); ^1^H-NMR: 2.26, 2.40, 2.46 (s, 9H, 3Me), 6.06 (s, 1H, Pyraz._(C4)_-H), 6.61 (brs, 2H, NH_2 Deutr. Exch_), 7.23 -7.36 (m, 4H, Thioph.-H and Pyrid._(C5)_-H), 9.61 (brs, 1H, NH _Deutr. Exch_); ^13^C-NMR: 10.7, 13.8, 38.3 (3Me), 113.7 (C≡N), 101.8, 106.8, 118.6, 124.3, 126.8, 127.2, 128.2, 128.6, 138.2, 138.6, 148.4, 153.3, 158.3, 162.9 (7C=C), 136.3, 155.7, 162.2, 162.8 (4C=N); MS (*m*/*z*, %): 431.11 (M^+^, 41); Anal. Calcd. for C_21_H_17_N_7_S_2_ (431.54): C, 58.45; H, 3.97; N, 22.72%. Found: C, 58.21; H, 3.69; N, 22.42%.

*Ethyl 3-amino-6-(2-(1,5-dimethyl-1H-pyrazol-3-yl)-4-methylthiazol-5-yl)-4-(thiophen-2-yl)-1H-pyrrolo[2,3-b]pyridine-2-carboxylate* (**10**). After recrystallization using dioxane, compound **11** was obtained as a pale yellow solid. Yield: 73%; m.p. 275–257 °C; IR (KBr): (cm^−^^1^) 675–681 (2C–S–C), 1612–1625 (3C=N), 1664 (C=O), 3320–3341 (NH and NH_2_); ^1^H-NMR: *δ* 1.29 (t, 3H, *J* = 6.5, Me), 2.42, 2.45, 3.64 (s, 9H, 3Me), 4.32 (q, 2H, *J* = 6.5, CH_2_), 6.01 (s, 1H, Pyraz._(C4)_-H), 6.32 (brs, 2H, NH_2 Deutr. Exch_), 6.83–7.49 (m, 4H, Thioph.-H and Pyrid._(C5)_-H), 10.62 (brs, 1H, NH _Deutr. Exch_); ^13^C-NMR: 10.8, 12.5, 14.1, 38.0 (4Me), 60.9 (CH_2_), 106.3, 108.3, 118.5, 121.5, 130.2, 127.6, 127.9, 128.0, 128.6, 138.2, 138.4, 147.5, 152.4, 154.0 (7C=C), 136.3, 151.1, 162.8 (C=N), 161.0 (C=O); MS (*m*/*z*, %): 478.12 (M^+^, 32); Anal. Calcd. for C_23_H_22_N_6_O_2_S_2_ (478.59): C, 57.72; H, 4.63; N, 17.56%. Found: C, 57.43; H, 4.51; N, 17.33%.

*7-(2-(1,5-Dimethyl-1H-pyrazol-3-yl)-4-methylthiazol-5-yl)-5-(thiophen-2-yl)pyrido[2,3-d]pyrimidin-4-amine* (**11**). The pyridine analogue **4** (0.39 g, 1 mmol) and formamide (10 mL) were fused face to face for 3 h. Afterwards, the separated brown solid on cooling was collected and dried. A brown solid with an 80% yield was obtained after recrystallization using EtOH; m.p. over 300 °C; IR (KBr): (cm^−^^1^) 675–681 (2C–S–C), 1610–1625 (5C=N), 3385 (NH_2_); ^1^H-NMR: *δ* 2.42, 2.45, 3.64 (s, 9H, 3Me), 6.01 (s, 1H, Pyraz._(C4)_-H), 6.32 (brs, 2H, NH_2 Deutr. Exch_), 6.83–7.49 (m, 5H, Thioph.-H, Pyrid._(C5)_-H and Pyrimi._(C2)_-H); ^13^C-NMR: 10.8, 12.5, 38.0 (3Me), 105.2, 106.3, 121.5, 127.7, 127.9, 128.0, 128.6, 138.2, 141.9, 144.9, 152.4, 157.8 (6C=C), 136.3, 151.2, 152.3, 157.4, 162.8 (5C=N); MS (*m*/*z*, %): 419.10 (M^+^, 9); Anal. Calcd. for C_20_H_17_N_7_S_2_ (419.53): C, 57.26; H, 4.08; N, 23.37%. Found: C, 57.11; H, 4.02; N, 23.17%.

*6-(2-(1,5-Dimethyl-1H-pyrazol-3-yl)-4-methylthiazol-5-yl)-4-(thiophen-2-yl)-1H-pyrazolo[3,4-b] pyridin-3-amine* (**12**). Compound **4** (0.39 g, 1 mmol) was refluxed for 3h with NH_2_OH.HCl (0.07 g, 0.1 mmol) in AcOH acid (25 mL) having anhydrous AcONa (0.08 g, 0.1 mmol) as a catalyst. Afterwards, the reaction mixture was decanted onto cold H_2_O, and the formed precipitate was filtered, dried and recrystallized from EtOH, affording a reddish-brown powder in a 91% yield; m.p. 286–288 °C; IR (KBr): (cm^−1^) 675–681 (2C–S–C), 1610–1625 (4C=N), 3320–3355 (NH and NH_2_); ^1^H-NMR: *δ* 2.42, 2.45, 3.64 (s, 9H, 3Me), 6.01 (s, 1H, Pyraz._(C4)_-H), 6.41 (brs, 2H, NH_2 Deutr. Exch_), 6.83–7.49 (m, 4H, Thioph.-H and Pyrid._(C5)_-H), 12.61 (brs, H, NH _Deutr. Exch_); MS (*m*/*z*, %): 407.10 (M^+^, 53); Anal. Calcd. for C_19_H_17_N_7_S_2_ (407.52): C, 56.00; H, 4.20; N, 24.06%. Found: C, 55.82; H, 4.11; N, 23.79%.

*General Method for the Synthesis of Compounds* (**13**–**16**). To a solution of the pyridine derivative **4** (0.39 g, 1 mmol), in AcOH acid (25 mL), equimolar amounts of some amino nucleophiles—namely urea, 6-methylpyridin-2-amine, 3-phenyl-1*H*-pyrazol-5-amine, and/or 1,2,4-triazin-3-amine—were added. Afterwards, the reaction mixture was refluxed for 6–8 h (Monitored by TLC analysis). After the solvent was evaporated in vacuo, the rough product was mushed with cold MeOH. The formed solid was isolated and then recrystallized to obtain compounds (**13**–**16**).

*4-Amino-7-(2-(1,5-dimethyl-1H-pyrazol-3-yl)-4-methylthiazol-5-yl)-5-(thiophen-2-yl)pyrido[2,3-d]pyrimidine-2(1H)-thione* (**13**). This was obtained as an orange-yellow solid after recrystallization using ethanol in a 72% yield; m.p. 197–199 °C; IR (KBr): (cm^−^^1^) 675–681 (2C–S–C), 1345 (thioamidic C=S), 1612–1625 (4C=N), 3210–3348 (NH, NH_2_); ^1^H-NMR: 2.26, 2.40, 2.46 (s, 9H, 3Me), 6.06 (s, 1H, Pyraz._(C4)_-H), 6.61 (brs, 2H, NH_2 Deutr. Exch_), 7.23–7.59 (m, 4H, Thioph.-H and Pyrid._(C5)_-H), 10.27 (brs, 1H, NH _Deutr. Exch_); ^13^C-NMR: 10.7, 13.8, 38.3 (3Me), 106.8, 118.6, 124.3, 127.2, 128.2, 128.6, 138.2, 138.6, 148.4, 153.3, 158.3, 162.9 (6C=C), 136.3, 155.7, 162.2, 162.8 (4C=N), 180.4 (C=S); MS (*m*/*z*, %): 451.07 (M^+^, 23); Anal. Calcd. for C_20_H_17_N_7_S_s_ (451.59): C, 53.19; H, 3.79; N, 21.71%. Found: C, 53.10; H, 3.51; N, 21.48%.

*2-(2-(1,5-Dimethyl-1H-pyrazol-3-yl)-4-methylthiazol-5-yl)-10-methyl-4-(thiophen-2-yl)-5H-dipyrido[1,2-a:3′,2′-e]pyrimidin-5-imine* (**14**). This was obtained as a yellow powder after recrystallization using dioxane in a 65% yield; m.p. 263–265 °C; IR (KBr): (cm^−^^1^) 675–6810 (2C–S–C), 1612–1625 (5C=N), 3219 (NH); ^1^H-NMR: *δ* 2.26, 2.42, 2.45, 3.64 (s, 12H, 4Me), 6.01 (s, 1H, Pyraz._(C4)_-H), 6.83–7.49 (m, 7H, Thioph.-H and Pyrid._(C3-5)_-H), 8.92 (brs, H, N=H _Deutr. Exch_); MS (*m*/*z*, %): 483.13 (M^+^, 41); Anal. Calcd. for C_25_H_21_N_7_S_s_ (483.61): C, 62.09; H, 4.38; N, 20.27%. Found: C, 61.79; H, 4.11; N, 20.10%.

*8-(2-(1,5-Dimethyl-1H-pyrazol-3-yl)-4-methylthiazol-5-yl)-2-phenyl-6-(thiophen-2-yl)pyrazolo [1,5-a]pyrido[3,2-e]pyrimidin-5-amine* (**15**). This was obtained as a yellow solid after recrystallization using MeOH/dioxane (3:1) in an 80% yield; m.p. 215–217 °C; IR (KBr): (cm^−1^) 675–681 (2C–S–C), 1610–1625 (5C=N), 3385 (NH_2_); ^1^H-NMR: *δ* 2.42, 2.45, 3.64 (s, 9H, 3Me), 6.01 (s, 1H, Pyraz._(C4)_-H), 6.52 (brs, 2H, NH_2 Deutr. Exch_), 6.83–7.49 (m, 11H, Ar-H, Thioph.-H, Pyrid._(C5)_-H and Pyrimi._(C2)_-H); ^13^C-NMR: 10.8, 12.5, 38.0 (3Me), 92.5, 105.2, 106.3, 121.7, 127.5, 127.6, 127.9, 128.0, 128.6, 128.7, 129.2, 133.0, 138.2, 141.9, 144.9, 149.4, 152.4, 159.2 (10C=C), 136.3, 152.3, 155.6, 157.4, 162.8 (5C=N); MS (*m*/*z*, %): 534.14 (M^+^, 17); Anal. Calcd. for C_28_H_22_N_8_S_2_ (534.66): C, 62.90; H, 4.15; N, 20.96%. Found: C, 62.71; H, 4.02; N, 20.69%.

*2-(2-(1,5-Dimethyl-1H-pyrazol-3-yl)-4-methylthiazol-5-yl)-4-(thiophen-2-yl)pyrido[3,2-e][1,2,4]triazolo[4,3-a]pyrimidin-5-amine* (**16**). This was obtained as faint yellow crystals after recrystallization using EtOH in a 75% yield; m.p. 218–220 °C; IR (KBr): (cm^−1^) 675–681 (2C–S–C), 1610–1625 (6C=N), 3385 (NH_2_); 2.26, 2.40, 2.46 (s, 9H, 3Me), 6.06 (s, 1H, Pyraz._(C4)_-H), 6.61 (brs, 2H, NH_2 Deutr. Exch_), 7.27–7.59 (m, 5H, Thioph.-H, Pyrid._(C5)_-H and Triaz._(C3)_-H); ^13^C-NMR: 10.7, 13.8, 38.3 (3Me), 106.8, 118.6, 124.3, 127.2, 128.2, 128.6, 138.2, 138.6, 148.4, 153.3, 158.3, 162.9 (6C=C), 136.0, 136.3, 155.3, 157.0, 162.2, 162.8 (6C=N); MS (*m*/*z*, %): 459.12 (M^+^, 55); Anal. Calcd. for C_21_H_17_N_9_S_2_ (459.55): C, 54.89; H, 3.73; N, 27.43%. Found: C, 54.59; H, 3.41; N, 27.22%.

### 3.4. Assessment of Anticancer Activity

#### MTT Cytotoxicity Assay

The in vitro growth inhibitory activity of the ten achieved analogues was explored in comparison with a notable anti-malignancy standard medication, etoposide, adapting the colorimetric MTT assay in triplicate as described previously [48]. In brief, cells were seeded onto 96-well tissue culture plates in DMEM containing 10% FBS to a final volume of 0.2 mL. The cells were subjected to different treatments after 24 h of seeding. The cells were then incubated for 48 h with etoposide (positive controls), test drugs or vehicle (DMSO). The media were then removed, replaced with 200 lL DMEM containing 0.5 mg/mL of MTT and cells were incubated for 2 h. Next, the supernatants were removed and the precipitated formazan was dissolved by adding 200 lL of DMSO. Absorbance at 570 nm was determined using a microplate reader (Model 450 Microplate Reader; Bio-Rad). Results were calculated by subtracting blank readings.

### 3.5. Docking Study

Topoisomerase II crystal structure with the cocrystallized ligand (etoposide) was downloaded from Protein Data Bank (PDB code: 3QX3). The target derivatives **5**–**13** and **16** were docked within the topoisomerase II active site using the MOE 2010 program. To validate the docking step, etoposide was redocked with RMSD = 0.9526. Three-dimensional structures of the target derivatives were built, protonated and energy minimized and saved as mdb files to be docked within topoisomerase II. The results of the docking study are given in Table 2.

### 3.6. Statistical Analysis

The presented results are mean ± SD, and the statistical analysis was performed using one-way ANOVA followed by Dunnett’s multiple comparisons test. Differences were considered significant at *p* < 0.05. Statistical analysis was performed using SPSS for Windows (SPSS, Inc., Chicago, IL, USA).

## 4. Conclusions

In this work, we described an efficient and facile approach for the synthesis of 2-amino-6-(2-(1,5-dimethyl-1*H*-pyrazol-3-yl)-4-methylthiazol-5-yl)-4-(thiophen-2-yl)nicotinonitrile (4), via one-pot multicomponent condensation, as a reactive precursor to synthesize novel pyrazolothiazole-based pyridine conjugates (**5**–**16**). All the target derivatives **5**–**13** and **16** were evaluated for their cytotoxic activity towards prostate cancer (PC-3), lung cancer (NCI-H460) and cervical cancer (Hela) cell lines. All the tested compounds revealed anticancer activity towards PC-3 (IC_50_ = 17.50–65.41 µM), NCI-H460 (IC_50_ = 15.42–61.05 µM) and Hela (IC_50_ = 14.62–59.24 µM). Analysis of the structure–activity relationship (SAR) indicated that the naphthyridine hybrids had more favorable cytotoxic potential than pyridopyrimidine, pyrrolopyridine and/or pyrido[3,2-*e*][1,2,4]triazolo[4,3-*a*] pyrimidine hybrids. A docking study was performed within the topoisomerase II active site to predict the binding mechanism of these compounds. All the docked compounds demonstrated good fitting within topoisomerase II. The most potent cytotoxic compound **7** (IC_50_ = 14.62–17.50 µM) displayed the best docking score (−17.29 Kcal/mol), forming two hydrogen bonds with the AspB479 amino acid, while the least potent cytotoxic compound **16** (59.24–65.41 µM) did not show any hydrogen bond with the enzyme and bound with DAC12 via hydrophobic binding with the lowest binding score (−10.98 Kcal/mol).

## Data Availability

The data presented in this study are available in this article.

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
