# Peer review of "Facile One-Pot Multicomponent Synthesis of Pyrazolo-Thiazole Substituted Pyridines with Potential Anti-Proliferative Activity: Synthesis, In Vitro and In Silico Studies"

_molecules, 2021, doi:10.3390/molecules26113103_

Round 1

Reviewer 1 Report

The work received for evaluation has as main topic a facile synthesis of a class of derivatives with potential anti proliferative activity. Although the manuscript shows a lot of organic chemistry work, there are some important issues that should be corrected/improved. The manuscript has 48 references written in the correct format.

-English must be checked; there are a lot of non common terms like line 52-'H bonding creation'; a lot of non-necessary ; as in 58,59,73 and so on. Line 81-'Expulsion of remaining solvents was accomplished by departure of the compartment'-???; line 83-'slim layer chromatography'; line 89 'smelting point', and many many others...

-line 48-figure 1 please remove bold;

-line 99-there is no details about docking study;

In short, the manuscript is difficult to be followed due to poor English and it is recommended to be edited by a native speaker with chemical knowledge.

Author Response

Dear Reviewer,

I would like to inform you that, our manuscript has been modified according to your valuable comments, as you will see throughout the manuscript.

Reviewing: 1

-English must be checked; there are a lot of non-common terms like line 52-'H bonding creation'; a lot of non-necessary; as in 58,59,73 and so on. Line 81-'Expulsion of remaining solvents was accomplished by departure of the compartment'-???; line 83-'slim layer chromatography'; line 89 'smelting point', and many many others...

First of all, we really thank the review for his/ her valuable comments, really we are sorry for these errors.

  • The English language was revised and improved throughout the whole manuscript, it rewritten by an English native speaker collage.
  • The style of the manuscript carefully revised, rewritten to match the reviewer’s comments.

-line 48-figure 1 please remove bold

  • It is Ok, and modified

-line 99-there is no details about docking study;

  • It was reported; page 7, Line 270

In short, the manuscript is difficult to be followed due to poor English and it is recommended to be edited by a native speaker with chemical knowledge.

  • Dear reviewer, really we would like to thank you for taking the time and efforts necessary to provide such insightful guidance. We thank the reviewer for her/his valuable comments regarding our article. We agree with all comments raised and found them very helpful. We went through the manuscript and modified the language in the revised version. In addition, the manuscript has been proofread by a native English speaker with expertise in the field.

Reviewer 2 Report

The manuscript molecules-1204761 "Facile one-pot multicomponent synthesis of pyrazolo-thiazole substituted pyridines with potential anti-proliferative activity: Synthesis, in vitro and in silico studies" by Islam H. El Azab et.al. describes the synthesis of  two series of 15 novel pyrazolothiazole–based pyridine conjugates and the study their anti-cancer activity.

Comments and remarks:

1) To date, there are a number of publications and reviews devoted to the subject of the manuscript (for example, reviews 10.1016/j.biopha.2021.111495, 10.3390/molecules23010134, papers 10.1016/j.bmcl.2017.08.003, 10.1007/s00044-017-1963-1, 10.1016/j.bioorg.2018.10.038 ect.). Authors need to clearly show the difference between their synthetic and biological results from published articles.

2) Since almost all compounds were obtained for the first time, new compounds should be characterized by few physical methods (1H, 13C NMR, IR spectroscopy and mass spectrometry). Images of all spectra should be added in supplementary materials.

3) Part "2.3 Molecular docking study" is missing. There are also problems with the numbering of sections.

4) Part 2.5.1. It is not clear what they used "doxorubicin or Erlotinib (positive controls)"

5) In how many replicates were all experiments performed? The results indicate that only in one. This is unacceptable for research work.

6) Why was etoposide chosen as the reference substance?

7) The study does not assess the impact of test compounds on normal cells.

Author Response

Dear Reviewer

I would like to inform you that, our manuscript has been modified according to tyour very valuable comments, as you will see throughout the manuscript.

Reviewer 2:

  • To date, there are a number of publications and reviews devoted to the subject of the manuscript (for example, reviews 10.1016/j.biopha.2021.111495, 10.3390/molecules23010134, papers 10.1016/j.bmcl.2017.08.003, 10.1007/s00044-017-1963-1, 10.1016/j.bioorg.2018.10.038 ect.). Authors need to clearly show the difference between their synthetic and biological results from published articles.
  • We really thank the review for his/ her time to review this manuscript, also, for his/ her very valuable comments. Out of the suggested works two refs. were already cited in our manuscript. Nearly, these suggested works and our manuscript all of them studied the same subject and showed very close outcomes. All disclosed the multicomponent synthesis and the high biological impacts of thiazole-containing scaffolds.
  • Since almost all compounds were obtained for the first time, new compounds should be characterized by few physical methods (1H, 13C NMR, IR spectroscopy and mass spectrometry). Images of all spectra should be added in supplementary materials.
  • We thank the reviewer for his/her time to evaluate our paper. All newly, synthesized compounds were fully characterized using MS, IR, 1H-NMR and Elemental analysis, but 13C-NMR was done for only some selected representatives compounds, so this analysis is so expensive and we cannot subject all newly synthesized compounds for all spectroscopic However, we reportedthe 13C NMR data for all new synthesized compounds except only, two compounds (12 and 14). We did our best to add the requested NMR data for the requested compounds. But, due to pandemic COVID-19 and limitations in working, all labs and analytical services are closed/limited working hours, we would appreciate your understanding.
  • Part "2.3 Molecular docking study" is missing. There are also problems with the numbering of sections.
  • It was reported; page 7, Line 270
  • Part 2.5.1. It is not clear what they used "doxorubicin or Erlotinib (positive controls)"
  • Really, we are so sorry for that, we used etoposide and was modified in the manuscript
  • In how many replicates were all experiments performed? The results indicate that only in one. This is unacceptable for research work.
  • Results are the average of three independent experiments run in triplicate.
  • Why was etoposide chosen as the reference substance?
  • Etoposide is a topoisomerase 2 (TOP2) inhibitor that demonstrated activity in patients with prostate cancer, lung cancer and ovarian cancer.

7) The study does not assess the impact of test compounds on normal cells.

  • The impact of the test compounds on normal cells will be assessed on another study, in due course.

Reviewer 3 Report

I really enjoyed reading this manuscript. From an organic point of view, this work can be used to learn mechanistic aspects related to relevant aza-heterocyclic systems. In addition, the results of the anticancer activity and docking molecular are relevant in medicinal chemistry and drug design. Unfortunately, I consider that manuscript could be accepted after major revision. This decision is owing to the absence of the Supplementary Material. In particular, 1H and 13C NMR spectra of each synthesized compound might be properly processed and included. Ultimately, some suggestions and comments are included: 

(1) See introduction, lines 74-77. It is very important to check this paragraph. It seems confuse.  

(2) See synthetic procedures and analytic data of compounds. (a) 1H-NMR (300 MHz, DMSO-d6): δ instead of 1H-NMR: δ, (b) for all proton signals appearing as doublet, triplet, among others, the corresponding coupling constant (J = xx.x Hz) might be included, (c) the scale ppm might be included at the end of each 1H and 13C NMR reporting data, (d) 13C-NMR (75 MHz, DMSO-d6): δ instead of 13C-NMR, (e) see line 106, I found  two signals at 2.42, 2.45 ppm; however, the multiplicity, coupling constant and/or integration is totally absent. Overall, the 1H and 13C NMR reporting data of each synthesized compound might be carefully revised.   

(3) See MTT cytotoxicity assay, lines 267-277. ¿Did you use a statistical analysis? ¿Did you use a duplicate of the measurements? ¿Did you determinate the standard deviation of the measurements?. For instance, it was included in the Table 1.

(4) See line 291. Pyridine analogues instead of pyridine analogies.

(5) See lines 295-296. Authors mention that a catalytic amount of AcONH4 was used. However, this starting material should be used in a stoichiometric amount (i.e. See lines 122-124).  

(6) See lines 296-297. Authors mention that “A plausible mechanism for the realization of 2-amino-3-cyano pyridine moiety 4 using NH4OAc, has displayed in Scheme 2”. From an organic point of view, the plausible mechanism might be properly explained.  

(7) See line 301, Scheme 1. pyran instead of pyan / analogues instead of analogies.

(8) See lines 316-317. Authors mention that “A reasonable mechanism for the formation of compound 8 was presented in scheme 4”. From an organic point of view, the plausible mechanism might be properly explained. The pathway to form compound 8 is very interesting.   

(9) See line 323. NH2 group instead of N-H2 groups. It is important to modify it in all sections.

(10) See Figures 3a, 4a, 5a, 6a, 7a, 8a, 10a, 11a, and 12a. Overall, the 3D binding mode within the active site. It is very important to improve the resolution and quality of them.  

(11) See references. Please, the DOI of each artichle might be included according to the guideline of the “Molecules”. 

Author Response

Dear Reviewer,

I would like to inform you that, our manuscript has been modified according to your valuable comments, as you will see throughout the manuscript.

Reviewer 3:

(1) See introduction, lines 74-77. It is very important to check this paragraph. It seems confuse.  

  • It is Ok, and modified

(2) See synthetic procedures and analytic data of compounds. (a) 1H-NMR (300 MHz, DMSO-d6): δ instead of 1H-NMR: δ, (b) for all proton signals appearing as doublet, triplet, among others, the corresponding coupling constant (J = xx.x Hz) might be included, (c) the scale ppm might be included at the end of each 1H and 13C NMR reporting data, (d) 13C-NMR (75 MHz, DMSO-d6): δ instead of 13C-NMR, (e) see line 106, I found two signals at 2.42, 2.45 ppm; however, the multiplicity, coupling constant and/or integration is totally absent. Overall, the 1H and 13C NMR reporting data of each synthesized compound might be carefully revised.   

  • We would like to thank the reviewer for taking the time and efforts necessary to provide such insightful guidance. We agree with all comments raised and found them very helpful.
  • We deleted the (300 MHz, DMSO-d6) and (75 MHz, DMSO-d6) so these data are reported in 2.2 Instrumentation page 3, lines 82-88. Really we did that to avoid the reputation and aiming to reduce the plagiarism. Hope you can understand us. Any way if you still need to add that we will do.
  • Dear reviewer all your notices already done as we will see throughout the manuscript.

(3) See MTT cytotoxicity assay, lines 267-277 ? Did you use a statistical analysis? Did you use a duplicate of the measurements? Did you determinate the standard deviation of the measurements? For instance, it was included in the Table 1.

  • Yes, we used statistical analysis and Results are the average of three independent experiments run in triplicate. All of these details were added in the manuscript and highlighted in yellow color.

(4) See line 291. Pyridine analogues instead of pyridine analogies.

  • It is Ok, and modified

(5) See lines 295-296. Authors mention that a catalytic amount of AcONH4 was used. However, this starting material should be used in a stoichiometric amount (i.e. See lines 122-124).  

  • Really, we are so sorry for this mistake, it is Ok, and modified

(6) See lines 296-297. Authors mention that “A plausible mechanism for the realization of 2-amino-3-cyano pyridine moiety 4 using NH4OAc, has displayed in Scheme 2”. From an organic point of view, the plausible mechanism might be properly explained.  

  • The plausible mechanism was explained in page 8, lines 292-295.  

(7) See line 301, Scheme 1. pyran instead of pyan / analogues instead of analogies.

  • It is Ok, and modified

(8) See lines 316-317. Authors mention that “A reasonable mechanism for the formation of compound 8 was presented in scheme 4”. From an organic point of view, the plausible mechanism might be properly explained. The pathway to form compound 8 is very interesting.   

  • The plausible mechanism was explained in pages 8 and 9, lines 314-320.  

(9) See line 323. NH2 group instead of N-H2 groups. It is important to modify it in all sections.

  • It is Ok, and modified in all sections

(10) See Figures 3a, 4a, 5a, 6a, 7a, 8a, 10a, 11a, and 12a. Overall, the 3D binding mode within the active site. It is very important to improve the resolution and quality of them.  

  • The quality and resolution of the above mentioned figures were improved.

(11) See references. Please, the DOI of each article might be included according to the guideline of the “Molecules”. 

All, the DOI of each article is included.

Round 2

Reviewer 1 Report

The authors improved the manuscript, but it still needs English adjustments. Although I am not a native English speaker, it seems not right. For example, the first sentences started with Lately, Conversely, Consequently, Recently...and some sentences are hard to be followed.

Author Response

We thank the reviewer so much for her/ his time to evaluate our paper for second round. Dear reviewer, swear you all of the requested spectra were already done and with our co-author Dr. Nadia, who was so sick and had a COVID-19 ten days, as we mentioned before, in an email to Ms. Rachel Liu; Assistant Editor (MDPI), on Wed, May 5 at 1:42 PM; before we receive the 2nd round reports.
Swear you my dear, she is in so critical case in the hospital and we cannot even communicate with her. So, we hope you can understand our so critical situation and forgive us of this request.
Dear Prof. all of us here really pray for the acceptance of that paper in the molecules, as well as, for our co-author dr. Nadia to come over her sick, thus, please, could you reconsider our manuscript for publication in molecules? Specially to pass a good news for dr. Nadia.

About the English language and style, we went through the manuscript and modified the language in the revised version. In addition, the manuscript has been proofread by a native English speaker with expertise in the field.

If the English language and style are still needs more improvement, we will subject our manuscript to molecules English editing service after acceptance.
Dear, really our situation her is so critical more than you can imagine, really all of us here are waiting your positive response
Thanks for help.

Reviewer 2 Report

This manuscript describes a great synthetic work. I believe that these physical methods should be confirmed by real images of spectra. Unfortunately, these data are not available for this manuscript. Therefore, I cannot evaluate the characterization of the obtained compounds.

Regarding the toxicity of the compounds obtained. Many heterocyclic compounds are toxic. The results of the high anticancer activity of any new compounds obtained are irrelevant if they are highly toxic.

Author Response

We thank the reviewer so much for her/ his time to evaluate our paper for second round. Dear reviewer, swear you all of the requested spectra were already done and with our co-author Dr. Nadia, who was so sick and had a COVID-19 ten days, as we mentioned before, in an email to Ms. Rachel Liu; Assistant Editor (MDPI), on Wed, May 5 at 1:42 PM; before we receive the 2nd round reports.
Swear you my dear, she is in so critical case in the hospital and we cannot even communicate with her. So, we hope you can understand our so critical situation and forgive us of this request.
Dear Prof. all of us here really pray for the acceptance of that paper in the molecules, as well as, for our co-author dr. Nadia to come over her sick, thus, please, could you reconsider our manuscript for publication in molecules? Specially to pass a good news for dr. Nadia.

  • Regarding the toxicity of the compounds obtained. Many heterocyclic compounds are toxic. The results of the high anticancer activity of any new compounds obtained are irrelevant if they are highly toxic.
  • The response, really we completely agree with that, therefor, the impact of the test compounds on normal cells will be assessed on another study, in due course. But, due to pandemic COVID-19 and limitations in working, all labs and analytical services are closed/limited working hours, we would appreciate your understanding.

  • About the English language and style, we went through the manuscript and modified the language in the revised version. In addition, the manuscript has been proofread by a native English speaker with expertise in the field. If the English language and style are still needs more improvement, we will subject our manuscript to molecules English editing service after acceptance

Dear, really our situation her is so critical more than you can imagine, really all of us here are waiting your positive response

Thanks for help.

Reviewer 3 Report

Although authors included most modifications suggested by reviewers, I follow considering that manuscript could be accepted after major revision owing to the complete absence of the Supplementary Material. In particular, 1H and 13C NMR spectra of each synthesized compound might be properly processed and included. This observation was included in the first review; however, it is not included in this version yet.   

Author Response

(The authors gave the same response as above.)
